# Design and Performance Verification of a Space Radiation Detection Sensor Based on Graphene

**DOI:** 10.3390/s21227753

**Published:** 2021-11-22

**Authors:** Heng An, Detian Li, Shengsheng Yang, Xuan Wen, Chenguang Zhang, Zhou Cao, Jun Wang

**Affiliations:** Science and Technology on Vacuum Technology and Physics Laboratory, Lanzhou Institute of Physics, Lanzhou 730000, China; ahllbl@126.com (H.A.); syang@sina.com (S.Y.); 15294156251@163.com (X.W.); zhangcg0319@163.com (C.Z.); caozhou-1@163.com (Z.C.); wangjunch1981@163.com (J.W.)

**Keywords:** space radiation detection, graphene, 2-D material, graphene field effect, radiation sensor

## Abstract

In order to verify the performance of a graphene-based space radiation detection sensor, the radiation detection principle based on two-dimensional graphene material was analyzed according to the band structure and electric field effect of graphene. The method of space radiation detection based on graphene was studied and then a new type of space radiation sensor samples with small volume, high resolution, and radiation-resistance was formed. Using protons and electrons, the electrical performance of GFET radiation sensor was verified. The designed graphene space radiation detection sensor is expected to be applied in the radiation environment monitoring of the space station and the moon, and can also achieve technological breakthroughs in pulsar navigation and other fields.

## 1. Introduction

High-resolution radiation detectors are able to distinguish narrow energy peaks near room temperature or at room temperature, providing new capabilities for radiation detection and measurement in areas such as material characterization, astrophysics, homeland security, and nuclear forensics. Studies have shown that silicon drifting detectors have been used to achieve a significant increase in energy resolution. There are also potential materials for high-resolution radiation detectors, such as high-density, high-charge materials, such as mercury iodide (HgI_2_), zinc, and cadmium (CdZnTe), etc. Although these materials have good energy deposition capacity and large-area preparation is also possible, their electrical properties and charge collection properties are affected by silicon and niobium crystals. In addition, these broadband compound semiconductors are far less mature and advanced than monomass semiconductors in doping, processing, and integrated circuit technology, thus limiting their further application. Table 1 shows the electrical properties of different semiconductor materials.

Graphene is a flat carbon atomic membrane structure material with unique electrical and material properties, including very high mechanical strength, very high bipolar migration rate and high thermal conductivity [1]. Graphene can be prepared by standard microchip manufacturing processes. Because graphene’s inherent capacitors are almost negligible, it is of particular application value in the development of high-resolution semiconductor radiation detectors at room temperature. Two-dimensional materials such as graphene have emerged as promising candidates for various electronic devices [2]. In recent years, graphene has been used to make various electronic devices and sensors [3]. Graphene is also developing into a 2D material that can be used in a wide range of industries, including as a medical sensor, radiation detector, etc. The integration of graphene into the field-effect transistor structure can provide a very sensitive reading mechanism for sensing carriers in semiconductor detectors, making it possible to prepare radiation-sensitive detectors with high resolution [4]. In addition, the combination of graphene, a substrate absorption layer and a neutron conversion layer can be used for neutron detection, and has the advantages of smaller size and higher conversion efficiency than plastic scintillation, and has broad application prospects in space neutrons, ground nuclear explosions, and space station neutron radiation environment detection, etc. Based on graphene’s energy belt structure and electric field effect, this paper mainly analyzes the irradiation response of electrons and protons for graphene field effect transistor, and provides guidance for the subsequent development of miniaturized detectors that can be used for space electron and proton detection.

## 2. Analysis of Detection Method of Graphene Field Effect Transistor

Graphene (XFNANO, Nanjing, China) is a zero-band gap semiconductor material with a conductor belt and a valence band that coincide at the Dirac point. Near the Dirac point, the number of graphene crystals in the k-space unit area is, using a dispersion relationship, expressed as:(1)$$N=\frac{g_{s}g_{v}\pi q^{2}}{{\left(2\pi \right)}^{2}}=\frac{g_{s}g_{v}E^{2}}{4\pi {\left(\hbar v_{F}\right)}^{2}}$$
where, *g_s_* = 2, *g_v_* = 2, are, respectively, spin degeneracy and energy-grain degeneracy. So, graphene’s state density near the Dirac point is:(2)$$D(E)=\frac{dN}{dE}=\frac{2\left|E\right|}{\pi {\left(\hbar v_{F}\right)}^{2}}$$

That is, graphene’s state density and energy at the Dirac point are also linear, and at *E* = 0, the state density is zero.

Under the action of an electric field, Dirac–Fermi can be continuously transformed from an electron (or hole) to a hole (or electron). Far from the Dirac point, graphene has only a single carrier, whose concentration is proportional to the loaded voltage. Because the conductivity is proportional to the carrier concentration, graphene’s resistance value is affected by voltage, as shown in Figure 1, which is the electric field effect of graphene.

Figure 1A above shows the basic structure of graphene field effect transistors [5]. Figure 1B shows the change of graphene resistance with the intensity of the electric field without irradiation, with the green circle in the figure showing the output resistance position of field-effect transistors. Figure 1C indicates that the graphene transistor has been exposed to radiation and that the substrate portion of the cell is ionized, and Figure 1D indicates that the output circuit of the graphene field effect transistor has shifted.

After the rays produce electrons in the detector’s absorption substrate—the cavity pair—under the action of gate voltage, the electron moves towards the graphene detection layer, and the hole moves towards the bottom of the substrate, detecting the number of electrons by bringing the electron drift below the graphene layer [6]. Using graphene’s energy belt structure and electric field effect, based on the characteristics of graphene Dirac point carrier concentration being extremely sensitive to the electric field, when the incident ray causes the absorption medium electric field to change, the graphene resistance value changes accordingly. Thus, the amount of change of graphene resistance value is used as the output signal of the detector to detect the space radiation.

Based on the above analysis, the graphene radiation sensor is designed to consist of three layers: a graphene sensitive detection layer, a SiO_2_ insulation layer and a Si semiconductor absorption substrate. Among them, the Si semiconductor absorbs the substrate as the working medium of radiation detection, absorbs the incoming rays and produces an electron–hole pair within the substrate; the SiO_2_ insulation layer in the graphene detection layer and Si Semiconductors absorb insulation between substrates, preventing electrons produced by radiation rays from being received directly by graphene. Graphene sensitive detection layers are mainly used to sense the electric field formed by radiation-producing electrons.

Ray incident to detector Si ionization in the substrate produces electrons called cavity pairs, and the number of resulting electron–cavity pairs is proportional to their energy. Load the voltage between the graphene detection layer and the Si semiconductor absorption substrate to form a suitable electric field distribution inside the detector, which directs the electrons (holes) generated by the radiation rays inside the substrate to drift towards the graphene detection layer and is blocked by the SiO_2_ insulation and eventually gathers below the graphene detection layer. The concentration of graphene carriers is regulated by the electric field by the concentration of the inductors of graphene, which changes the resistance value of graphene by the electric field in which the inductive electrons (holes) are concentrated [7]. By measuring the amount of change in graphene resistance value, it is possible to calculate the amount of change in the intensity of its electric field [8], and thus the number of electron-hole pairs produced by incoming and exiting radiation rays can be deduced, in conjunction with the subsequent electronic system to obtain the amount of injection of incident rays as shown in Figure 2.

As can be seen from Figure 2, the geometry of the device is similar to that of a metal–oxide–metal (MOS) structure. When the voltage is applied to the semiconductor through the gate contacted by ohm on the back, an exhausting zone is formed below graphene in the semiconductor. In the exhaust zone, the remaining fixed charge produces an internal electric field that separates the electron–hole pairs produced by the impact of energy particles. Separated charges are aggregated at the insulator/semiconductor interface, enhancing the lateral electric field applied to graphene. By applying a constant source of leakage voltage, changes in graphene conductivity can be measured as changes in leakage current. The depth of the depletion area determines the number of electron–hole pairs to be collected and is related to the density of the doping agent and the size of the back gate voltage.

## 3. Simulation Analysis

The penetration path and energy deposition of high-energy electrons are simulated by Monte Carlo simulation, mainly to obtain the penetration depth and energy deposition of incident rays in the substrate medium [5]. Taking 500 keV electrons as an example of how rays interact with substrate media, Figure 3 simulates the trajectory of electrons with 500 keV energy in semiconductor silicon materials. The red line in Figure 3 represents the trajectory of the electrons that escape after backscattering, the blue lines represent the trajectory of the electrons in the Si, and the maximum penetration depth of the 500 keV electrons can reach 680 m. In the actual sensor preparation process, the thickness of the silicon absorption layer can be designed according to the energy range of the detected particles [9]. The depth of penetration and ionizing energy loss of 5 MeV protons in silicon materials is shown in Figure 4.

## 4. Test Device Preparation

Using the CVD method to deposit single-layer polycrystalline graphene film, the test device is prepared, which is shown in Figure 5.

In order to detect space particles, semiconductor detection sensitive area can be measured in millimeters, the using of etching technology to etch graphene film into an array distribution of micron-sized graphene units, made of graphene devices. The thickness of Si substrate is 700 μm, and the thickness of the SiO_2_ insulation layer is 200 μm. The thickness of the electrode with Au is 40 nm. The monolayer graphene sensitive area is 10 μm × 10 μm. Figure 6 below shows the cross-section of the graphene layers, source poles, and gates shown in Figure 2.

Figure 6 shows the distribution of electrodes resulting from annealing of the graphene field-effect transistor devices made to improve the contact characteristics of the electrodes^,^ so that they can be as much as possible ohm contact.

In order to characterize the test, the process of gold wire lead welding is required to lead out the electrodes of the micro-devices [10], so as to facilitate the package interconnection of the devices, as shown in Figure 7 of the device with gold wire leads, and the electrodes 1 and 2 contain leads [11].

## 5. Test Results and Analysis

### 5.1. Experiments and Data

The electrical properties of graphene devices are verified by electron and proton irradiation. The parameters for the electron irradiation test are shown in Table 2 below. After the sample preparation is complete and the initial Raman test is carried out, the sample to be irradiated is vacuum-encapsulated, the sample is removed during irradiation, and the sample is vacuum-sealed again after irradiation is completed [12], pending subsequent testing.

Figure 8 shows the curve of the change of output characteristics before and after electronic irradiation of graphene devices. The applied source leakage bias voltage V_DS_s = 1.0 V, the gate voltage V_GS_s = 0 V, and the electrical characteristics test are tested by the semiconductor parameter test analyzer Keysight B1500A measured. In order to ensure the comparability of electrical characteristics between the devices, the electrical characteristics test data after irradiation are normalized.

Figure 8A is the output characteristic curve under linear coordinates, where the solid line is the test result before the electron irradiation and the dashed line is the test result after the electron irradiation. It can be seen that after different doses of electronic irradiation, the output characteristics of graphene devices decreased significantly, indicating that electronic irradiation has a greater impact on the electrical characteristics of graphene devices. In order to compare the degree of influence of different irradiated dose samples, the linear coordinates (*Y*-axis) after the comparable coordinate processing can be obtained from Figure 8B. The output characteristics of graphene samples decreased by about one to two magnitudes, but the degree of decline in output characteristics did not show a clear trend with the increase in the amount of electron irradiation.

Figure 9 is the transfer characteristic curve before and after irradiation of graphene devices, where Figure 9A is linear and Figure 9B is the logarithmic coordinate. The gate pressure applied during the test is V_GS_ = −50 to 50 V, and the source leak bias V_DS_ is 0.1 V. It can be seen that the current of graphene devices after electronic irradiation decreased significantly, and the numerical treatment was carried out to compare the degree of decline of different irradiated doses. As can be seen from Figure 9B, the degree to which the normalized current after electron irradiation decreases with the increase in gate pressure is less than the magnitude before electron irradiation, which indicates that the device’s gate controls the concentration of graphene channel carriers.

For proton irradiation experiments, the irradiation energy of 40 keV is also set, and the proton irradiation test parameters are shown in Table 3 below. Graphene samples are stored in the same way as electron irradiated samples.

Figure 10 shows the output characteristic curve of graphene devices before and after proton irradiation, Figure 10A shows the linear coordinates, and Figure 10B shows the regressive coordinates, where the solid line is the output curve before proton irradiation and the dotted line is the output curve after proton irradiation. The test parameters in the test are the same as the output characteristics test parameters of the electronic irradiated device. As can be seen from Figure 10A, proton irradiation has a significant impact on the output characteristics of graphene devices, resulting in a significant reduction in the source leakage current, and as can be seen from Figure 10B, as the amount of irradiated injection increases, the source leakage current of the sample decreases gradually, which indicates that the proton irradiation dose is proportional to the degree of influence of the electrical properties of the device.

Figure 11 shows the transfer characteristic curve of graphene devices before and after proton irradiation, and the test parameters of transfer characteristics are the same as those of electronic irradiated devices. It can be seen that as the proton irradiation dose increases, the source leakage current decreases in turn, which indicates that proton irradiation is more destructive to graphene. In addition, it can be seen from the transfer characteristic curve that after proton irradiation, the normalized source leakage current decreases with the increase in gate voltage, which indicates that the gate control capacity of the graphene device decreases, which is related to the defect of proton irradiation and the damage to the SiO_2_ media layer.

Figure 8 and Figure 10 show the electrical response of G-FETs upon the electron and proton irradiation respectively. The results in Figure 8 suggest that the current change of the G-FET was about 0.60 at 0 V gate voltage after the proton irradiation. However, the current change reported in Figure 10 was as low as 0.50 at 0 V gate voltage after proton irradiation. This inconsistency could be caused by electrons penetrating deeper than protons. Perhaps the deeper the electron penetrates, the more hole pairs it induces.

### 5.2. Results Analysis

The current stability test curve (I–t curve) for graphene samples before and after electron irradiation is shown in Figure 12 below, where the solid line is the I–t curve before irradiation and the dotted line is the I–t curve after irradiation. The source leakage voltage applied during the test is V_DS_s = 0.1 V, gate voltage V_GS_s = 0 V, and the test time is 60 s. From this diagram, we can see the change of device source leakage current over time, so as to determine the stability of the device. It can be seen from Figure 12 that the drop in current of the graphene device after irradiation is obvious. In addition, the current stability of the graphene device after electron irradiation is slightly worse, of which the stability of 2–4 is relatively serious, but overall, the graphene device is still relatively stable after irradiation [13].

After extracting the leakage current at the source leakage voltages V_DS_ = 0.1 V and V_DS_ = 1.0 V, and then normalizing them separately, the relationship between the leakage current and the electron irradiation dose can be obtained, as shown in Figure 13. It can be seen from Figure 13 that electron irradiation has a significant effect on the electrical properties of graphene. With the increase in irradiated dose, the leakage current shows a tendency to decrease significantly and then increase. Comparing the leakage current at source leakage voltage V_DS_s = 0.1 V and V_DS_s = 1.0 V, it can be seen that the leakage current at V_DS_s 0.1 V is more affected by electron irradiation than the leakage current at V_DS_s = 0.1 V, but the trend of the two is the same.

The source leakage current stability test curve of graphene samples before and after proton irradiation is shown in Figure 14, where the solid line is the I–t curve before irradiation and the dotted line is the I–t curve after irradiation. The test parameters are the same as for electronic irradiated devices. It can be seen that although the source leakage current decreased significantly after proton irradiation, the stability of the current is still good, indicating that proton irradiation does not affect the stability of device I-V characteristics.

After the leakage current is normalized to extract the leakage current at the source leakage voltages V_DS_ = 0.1 V and V_DS_ = 1.0 V, the leakage current can be obtained with the change of proton irradiation dose, as shown in Figure 15. It can be seen in Figure 15 that proton irradiation on graphene samples is equally large, with the increase in proton irradiation dose, and the decrease in normalized leakage current. The leakage current at 1.0 V is the same as the downward trend and the degree of decline. In addition, comparing the effects of electron irradiation with that of proton irradiation, it can be seen that the effect of proton irradiation is slightly greater than that of electron irradiation.

## 6. Conclusions

The application of a graphene field effect transistor (GFET) in collecting radiation-induced charge was verified. The experimental study found that, without any protection, the device electrical and photoelectric performance attenuation is small, showing a strong radiation protection ability. If the device is subsequently given packaged protection, the anti-radiation characteristics of the device will be further enhanced, making the impact of radiation damage smaller. However, in electronic and proton irradiation, it is also found that the electro-transmission characteristics of graphene devices are relatively large, so in practical applications, the device needs to have some package and other protection, so as not to expose the device to high doses of irradiation [14]. The results show that graphene can be transferred to a semiconductor substrate with different strands and can be used to make functional GFET devices using these sediments. In addition, it has been found that the detection of photoelectric effects and radiation-induced carriers can be achieved by changing the current flowing through the GFET device, which is dependent on voltage.

## Figures and Tables

**Figure 1 sensors-21-07753-f001:**
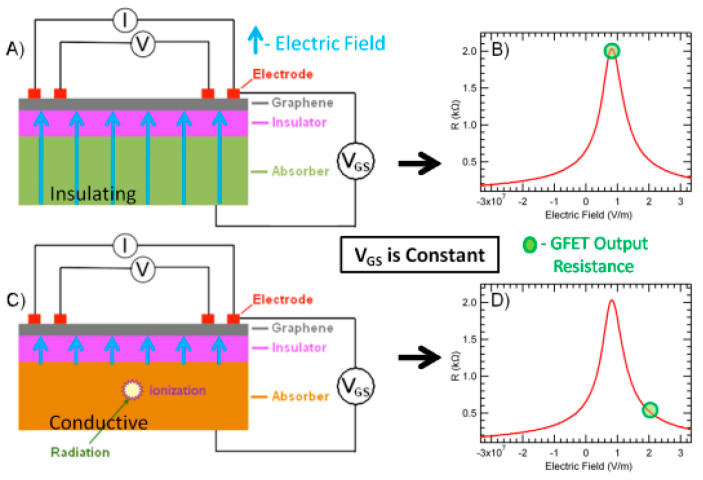
The electric field effect of graphene. (**A**) the basic structure of graphene field effect transistors [5]. (**B**) the change of graphene resistance with the intensity of the electric field without irradiation, with the green circle in the figure showing the output resistance position of field-effect transistors. (**C**) the graphene transistor has been exposed to radia-tion and that the substrate portion of the cell is ionized. (**D**) the output circuit of the graphene field effect transistor has shifted.

**Figure 2 sensors-21-07753-f002:**
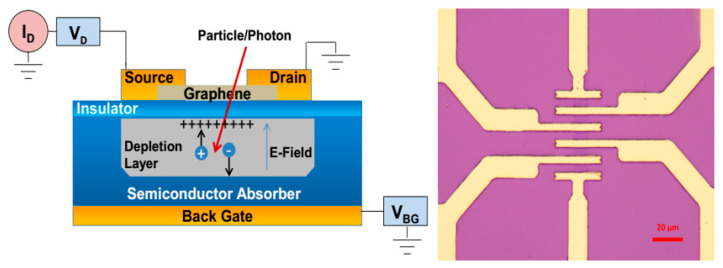
How graphene space radiation detectors work.

**Figure 3 sensors-21-07753-f003:**
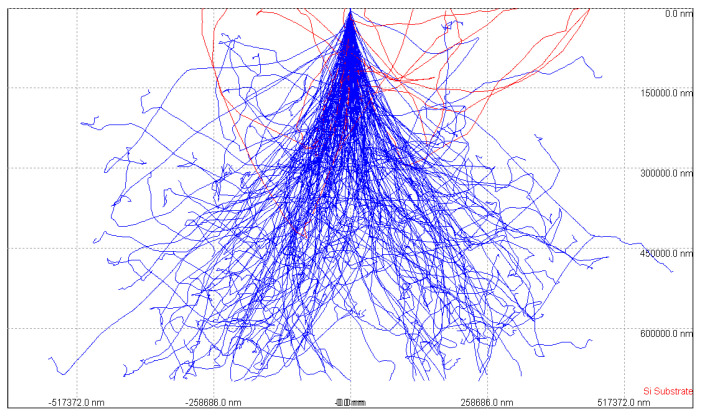
500 keV electron trajectory in Si.

**Figure 4 sensors-21-07753-f004:**
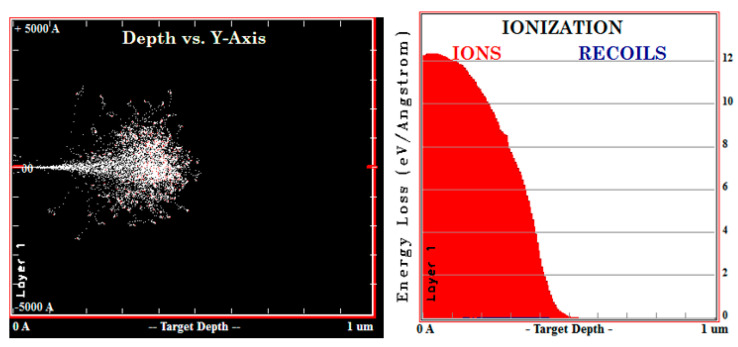
Penetration depth and ionization energy loss of 40 KeV and 5 MeV proton.

**Figure 5 sensors-21-07753-f005:**
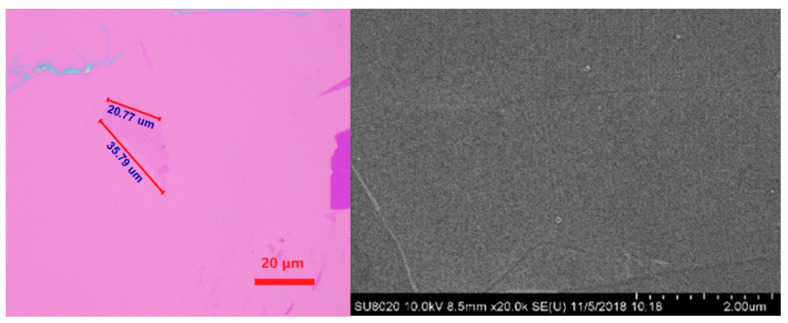
CVD graphene and SEM image.

**Figure 6 sensors-21-07753-f006:**
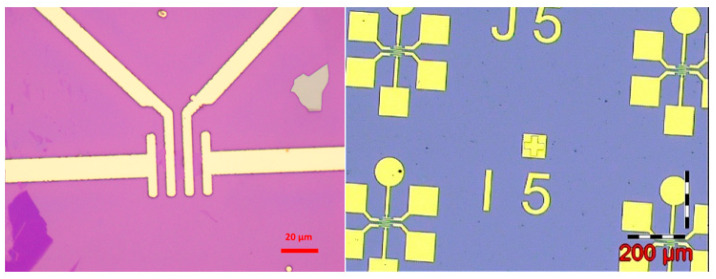
Graphene electrode pattern and graphene device channel diagram.

**Figure 7 sensors-21-07753-f007:**
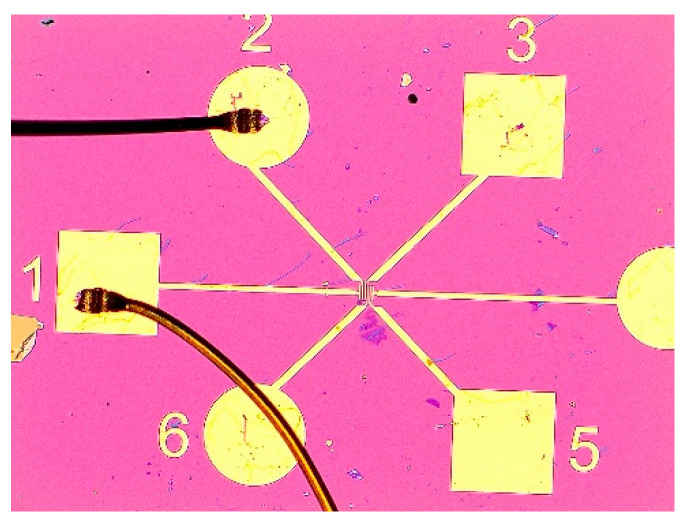
Schematic diagram of gold wire welding graphene device.

**Figure 8 sensors-21-07753-f008:**
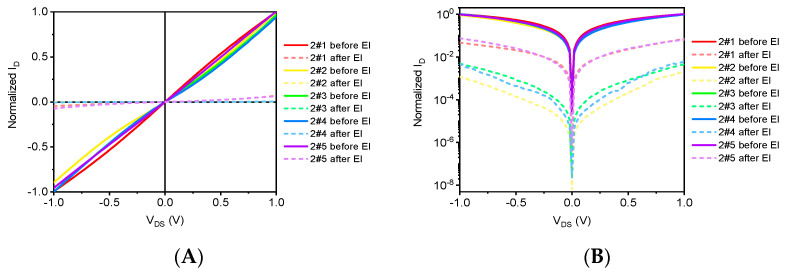
The output characteristic curve of graphene samples before and after electron irradiation. (**A**) Linear coordinates and (**B**) Logarithmic coordinates.

**Figure 9 sensors-21-07753-f009:**
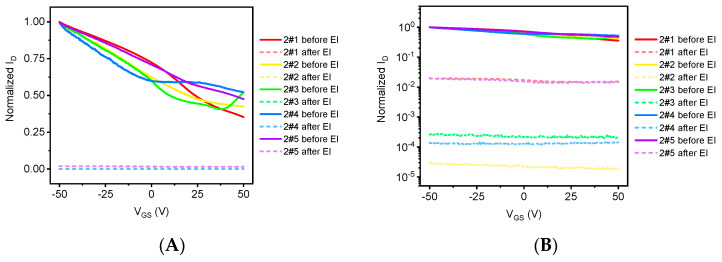
The transfer characteristic curve of graphene samples before and after electron irradiation. (**A**) Linear coordinates and (**B**) Logarithmic coordinates.

**Figure 10 sensors-21-07753-f010:**
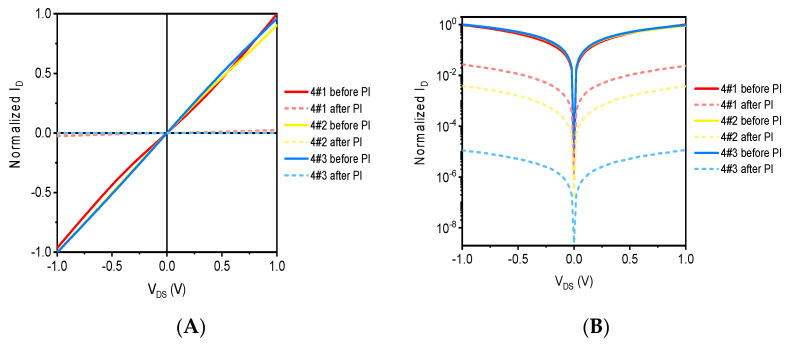
The output characteristic curve of graphene samples before and after proton irradiation. (**A**) Linear coordinates and (**B**) logarithmic coordinates.

**Figure 11 sensors-21-07753-f011:**
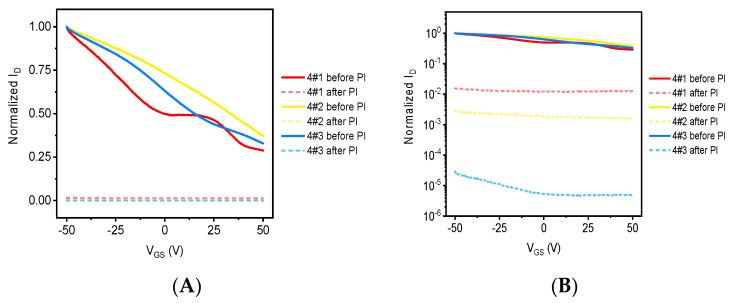
The transfer characteristic curve of graphene samples before and after proton irradiation. (**A**) Linear coordinates and (**B**) logarithmic coordinates.

**Figure 12 sensors-21-07753-f012:**
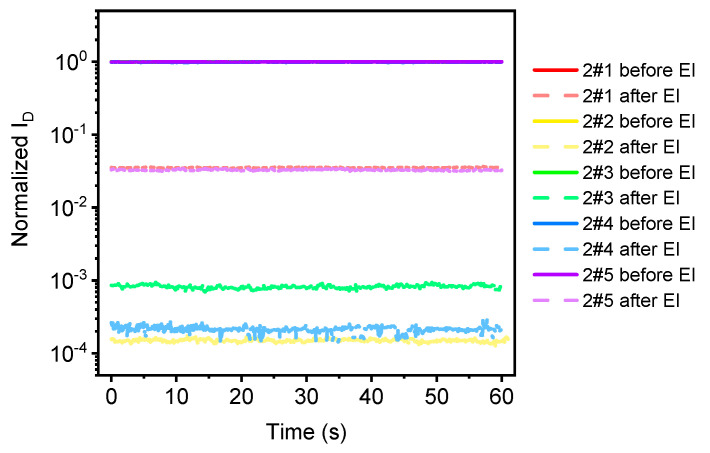
I–t curve of graphene samples before and after electron irradiation.

**Figure 13 sensors-21-07753-f013:**
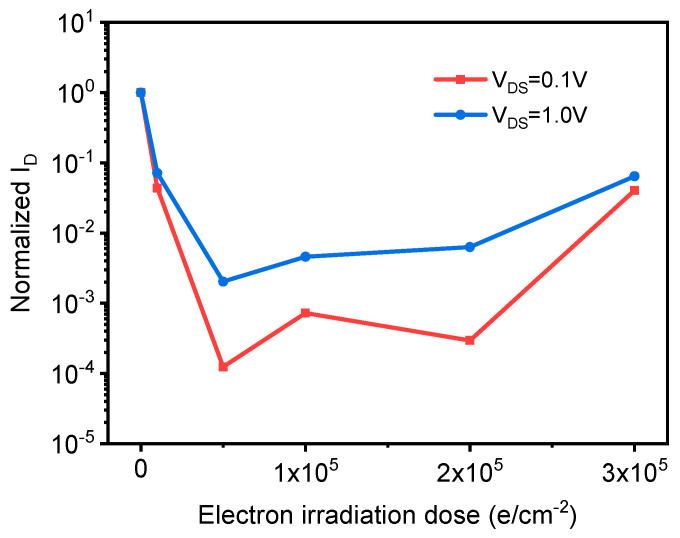
The normalized leakage current of graphene samples varies with the dose of electron irradiation.

**Figure 14 sensors-21-07753-f014:**
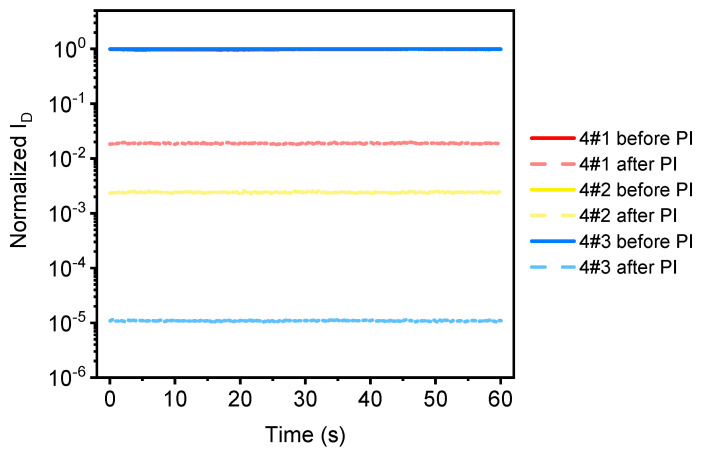
The I–t curve of the graphene sample before and after proton irradiation.

**Figure 15 sensors-21-07753-f015:**
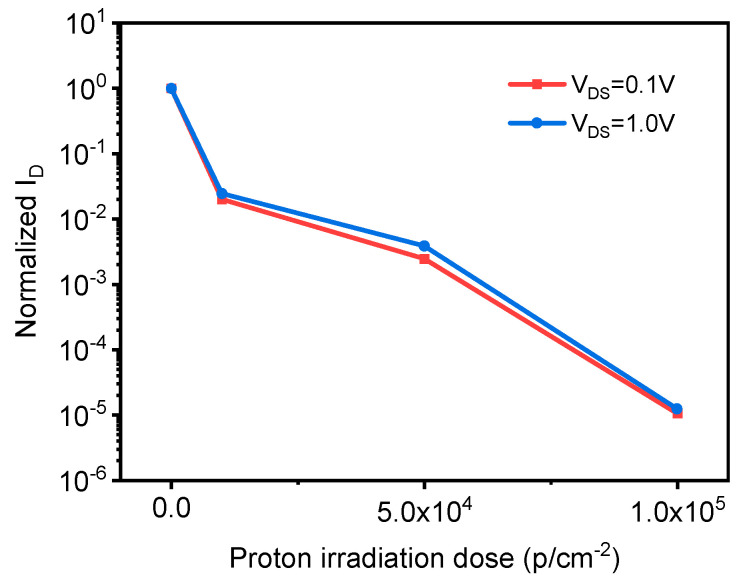
The normalized leakage current of graphene samples varies with proton irradiation dose.

**Table 1 sensors-21-07753-t001:** Characteristics of semiconductor materials for radiation detection.

Mat	Density/g/cm^3^	Band Gap/eV	Electron Mobility/cm^2^/V·s	Hole Mobility/cm^2^/V·s	Ratio of Mobility
Si	2.33	1.12	1900	500	3.8
6H-SiC	3.21	2.86	400	75	5.3
Ge	5.33	0.67	3800	1820	2.1
HgI_2_	6.4	2.13	100	4	25
CdZnTe	6.0	1.64	1350	120	11.3

**Table 2 sensors-21-07753-t002:** Electron irradiation experimental parameters.

Irradiation Type	Electron Irradiation
Irradiated Energy (keV)	40
Irradiation injection (e/cm^2).^	1 × 10^4^, 5 × 10^4^, 1 × 10^5^, 2 × 10^5^, 3 × 10^5^

**Table 3 sensors-21-07753-t003:** Proton irradiation experimental parameters and sample classification.

Irradiation Type	Proton Irradiation
Irradiated Energy (keV)	40
Irradiation injection (p/cm^2).^	1 × 10^4^, 5 × 10^4^, 1 × 10^5^

## Data Availability

The data presented in this study are available on request from the corresponding author.

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
