# Peer review of "Design and Performance Verification of a Space Radiation Detection Sensor Based on Graphene"

_sensors, 2021, doi:10.3390/s21227753_

Round 1
Reviewer 1 Report
The authors designed and tested a new radiation detection sensor based on graphene. The test results show that it may be used as a space radiation detector due to its excellent radiation detection and protection ability.
The designed radiation detection sensor based on graphene consists of graphene layer, SiO2 insulation layer and Si semiconductor absorption substrate. It is better to specify the dimension of the three layers in section 3 ‘Test device preparation’ instead of using words of millimeters and micron-sized.
The depth of penetration in silicon (or range) with 40 keV proton is suggested to be calculated because the proton beam energy for test is 40 keV (Table 2).
Author Response
Revised one by one according to the opinions of the reviewer,please see the attachment.

Reviewer 2 Report
The authors describe a graphene-based radiation detection sensor device. The paper is well written, and the results are very comprehensive. Also, the conclusion is convincing. Hence the paper is recommended to be accepted for publication. Prior to that, some issues need to be addressed.
(1) The G-FET is fabricated by using the CVD-grown graphene. Can the authors show the optical images of G-FET device at Fig.2 ?
(2) In Fig. 10, the transfer curves of the developed G-FET show a p-type behavior. However, it is known that the G-FET is n-typed in most cases. Can the authors explain why their device is p-typed.
(3) In Fig.10, it is obvious that Diac point current not clearly observed at zero. While the current decreases along with the increased VG = -0.1 V. Why, Is there any scientific reason?
(4) Figure 8 and 10 show the electrical response of G-FETs upon the proton irradiation. The results in Figure 8 suggest that the current change of the G-FET was about 0.60 at 0 V gate voltage after the proton irradiation. However, the current change reported in Figure 10 was as lower as 0.50 at 0 V gate voltage after proton irradiation. Can the authors explain the inconsistency between the two figures?
(5) The graphene-based radiation detection sensor's sensitivity and limitations should be more clearly reported in the introduction by giving a table. GFET is now utilized to detect ionizing radiation, and the author should explain why this method is preferable to other options. What are the advantages of such a sensor over the present radiation detection technologies.
(6) The reviewers recommend to discuss some relevant applications of 2D material content to improve this manuscript's readability. For example, 2D materials such as graphene has emerged as promising candidate for various electronic devices (Nature 516, 227-230 (2014) and Adv. Electron. Mater. 2020, 6, 1901100).
Author Response

(The authors gave the same response as above.)
